# Effect of a Combination of Ultrasonic Germination and Fermentation Processes on the Antioxidant Activity and γ-Aminobutyric Acid Content of Food Ingredients

Natalya Naumenko [1,*] , Rinat Fatkullin [1], Natalia Popova [1], Alena Ruskina [1], Irina Kalinina [1] ,
Roman Morozov [2], Vyacheslav V. Avdin [2], Anastasia Antonova [3] and Elizaveta Vasileva [4]

[1] Department of Food and Biotechnology, South Ural State University (National Research University),
76 Lenin Avenue, Chelyabinsk 454080, Russia
[2] Scientific and Educational Center "Nanotechnologies", South Ural State University (National Research University),
76 Lenin Avenue, Chelyabinsk 454080, Russia
[3] Faculty of Biotechnology, ITMO University, Kronverksky Prospect, 49, A., St. Petersburg 197101, Russia
[4] Department of Digital Technologies for Transport Process Management, RTH (MIIT), Obraztsova st., 9,
Building 9, Moscow 127994, Russia
* Correspondence: naumenkonv@susu.ru; Tel.: +7-91-9312-2375

**Abstract:** Whole-grain food ingredients enable the most balanced food products to be obtained, thus forming an important part of a healthy and sustainable diet. Wheat and barley grains are a traditional source of food ingredients for breads, breakfasts, drinks, and snacks in Russia. Such foods are suitable for all ages with many health benefits. However, the modern metropolitan citizen consumes large quantities of refined cereal products, thus impoverishing their diet. An alternative in dietary fortification could be sprouted and fermented food ingredients with an increased nutritional value. The present work was carried out to study the effect of a combination of germination with ultrasound treatment and fermentation with a complex starter of cereal crops on antioxidant activity and γ-aminobutyric acid content of food ingredients with the possibility of using them in the matrix of food products. In order to obtain germinated food ingredients, we used crops with the highest yield in the Ural region (Russia): two samples of soft spring wheat (*Triticum aestivum* L.) and a sample of spring barley grain (*Hordeum vulgare* L.). Obtaining food ingredients was divided into successive stages: ultrasonic treatment ($22 \pm 1.25$ kHz) was performed by means of changing power and length of time (245 W/L, intensity for 5 min); germination and fermentation used complex starter "Vivo Probio". The proposed technology of germination with haunting fermentation of cereal crops resulted in food ingredients with a more uniform distribution of granulometric composition, a low proportion of fine particles (4.62–104.60 μm) ($p < 0.05$) and large particles (418.60–592.00 μm) ($p < 0.05$). The particle size range (31.11–248.90 μm) ($p < 0.05$) was predominant. The germination and fermentation process resulted in 26 to 57% ($p < 0.05$) lower phytic acid content, 35 to 68% ($p < 0.05$) higher flavonoid content, 31 to 51% ($p < 0.05$) higher total antioxidant activity, 42.4 to 93.9% ($p < 0.05$) higher assimilability, and 3.1 to 4.7 times ($p < 0.05$) higher γ-aminobutyric acid content, which will allow production of food products with pronounced preventive action. The data was analyzed via one-way ANOVA analysis of variance using the free web-based software. The combination of the germination process with ultrasound treatment and subsequent fermentation with a complex starter can be used to support the development of healthful food products with increased GABA and antioxidant activity.

**Keywords:** grain crops; flavonoids; complex starter; nutritional enhancement; whole grains

## 1. Introduction

Growing consumer demand for whole-grain food ingredients and high-quality food products has spurred the development of research and innovative technologies for creating

food products with improved natural flavours, combined with a reduction in the use of enhancers and fortifiers of various chemical natures [1–3].

Whole-grain food ingredients contain a unique phytochemical complex: phenolic acids, anthocyanidins, quinones, flavonols, chalcones, flavones, flavanones and aminophenolic compounds [4]. Sprouted cereal crops have a higher nutritional and physiological value than their processed products [5]. Germination is a biochemical process which, when carried out under controlled conditions, effectively improves nutrients [6,7].

In recent years, the germination of cereal crops has been actively used to obtain new food components used in the manufacture of products. This is due to an increase in their nutritional value and improved absorption of nutrients of the ingredients created [8]. Cereal products obtained using sprouted raw materials have been reported to have a better flavour, a softer consistency, and a distinctly sweet taste [9]. Both the germination process and the fermentation process are known to activate enzymes, thus helping to increase the digestibility of the grain. Grain germination as well as the fermentation process increases the availability of reducing sugars free amino acids including lysine [10,11] and stimulates the accumulation of $\gamma$-aminobutyric acid (GABA) [12,13], minerals [14], dietary fibre [15], and phenolic compounds and increases antioxidant activity [16].

Ultrasound, as a new and promising non-thermal processing technology, is actively used as an industrial tool in the food industry (in the frequency range from 20 to 100 kHz) [17]. Studies by M. Boukroufa describe the favorable effect of ultrasound on the content of flavonoids in plant raw materials [18]. Junzhou Ding et al. showed that the application of ultrasound treatment (25 kHz) for 5 min allows the enrichment of plant raw materials metabolites, including GABA and riboflavin (vitamin B 2).

Our earlier studies [19] proved that the use of ultrasound (20 kHz) with 227 W/L for 3 min during germination of wheat grain *Triticum aestivum* L. enables intensification of the germination process, increases antioxidant activity (2.86 mg/g Trolox equivalents) and flavonoid amounts (0.19 mg QE/g) and activates GABA synthesis (18.9 mg/100 g).

Researchers [20] also note that the content of GABA increases in plant raw materials when using ultrasound treatment (40 kHz, 300 W) during the process of soaking for 30 min. Other researchers have shown that ultrasound treatment (25 kHz, 5.1 W/L) for 10–30 min can activate the accumulation of vitamins B 1, B 2, and B 3 [21]; treatment (25 kHz, 26 W/L) increases antioxidant capacity [22].

GABA is an amino acid free of four carbon atoms. It is based on the irreversible decarboxylation of glutamic acid by glutamate decarboxylase [23,24]. GABA is a neurotransmitter in the nervous system which can improve cerebral blood flow and promote brain cell metabolism. It is known to be an excellent substance for the treatment of neurological diseases; it also has the ability to lower blood pressure, has diuretic effects, improves liver function, and stimulates alcohol metabolism [25,26].

GABA is widespread in plant foods but is found in very low concentrations, as in the original wheat grain 1–4 mg/100 g in white rice, 4–8 mg/100 g in brown rice, and 10–100 mg/100 g. Germination processes can increase these values to concentrations of 15 to 18 mg/100 g, but this is not sufficient for a sustained preventive effect.

Another way to increase the GABA content of natural plant material is fermentation. In a recent study using Saccharomyces cerevisiae, a method of producing a functional fermented apple drink with increased GABA content by fermentation of apple juice was reported [27].

The authors [28] noted in their studies that lactic acid bacteria can efficiently produce GABA from glutamic acid. Park and Oh and Han et al. presented a method to increase GABA content by fermenting soybeans to produce plant-based beverages [29,30].

However, studies combining the effects of ultrasound exposure before the germination process with subsequent fermentation of cereal crops on antioxidant properties, GABA content, and other characteristics of food ingredients are still not well understood in the literature and are thus of scientific interest.

The aim of this study was to study the combination of the processes of germination with ultrasound treatment and fermentation of complex starter cereal crops on the antioxidant activity and GABA content of the obtained raw ingredients with the possibility of using them in the matrix of food products.

## 2. Materials and Methods

### 2.1. Materials

The crops used in this study were the crops with the largest yield in the Ural region (Russia) between 2018–2022. They are most often stored and used in technologies for obtaining raw ingredients:

- grain of soft spring white wheat (*Triticum aestivum* L.), variety Zauralochka, harvested in 2022, grown in the Ural region, Russia. The protein content was 12.5 ± 2.1 g/100 g in terms of moisture content [31];
- grain of soft red spring wheat (*Triticum aestivum* L.), variety Erythrosperium, 2022, grown in the Ural region, Russia. The protein content was 15.5 ± 1.8 g/100 g in terms of moisture content [32];
- spring barley grain (*Hordeum vulgare* L.), variety Chelyabinets 1, crop 2022, grown in the Urals region, Russia. The protein content was 13.2 ± 1.9 g/100 g in terms of moisture content [33].

All samples were grown in the steppe zone of the Bredinsky municipal district of the Chelyabinsk region of the Russian Federation, which belongs to the zone of critical farming. Grain sampling was carried out in accordance with the requirements of GOST 13586.3-2015.

The grain crops used were pre-sampled and aligned by length and width using SeedCounter v.1.9.5 grain phenotyping software developed by employees of Novosibirsk State University (Novosibirsk, Russia) [34]. Grain phenotyping criteria were defined for wheat grain: length 6–7 mm and width 2.1–2.3 mm; for barley grain, length 8–10 mm and width 2.5–2.7 mm.

The objects of the study were defined as:

A. food ingredient obtained by milling the whole grain of soft spring white wheat of the Zauralochka variety before germination and fermentation;
B. food ingredient obtained by milling whole red soft spring wheat of the variety Erythrosperium before germination and fermentation;
C. food ingredient obtained by milling whole spring barley Chelyabinets 1 before germination and fermentation;
D. food ingredient obtained by milling germinated and fermented whole spring white wheat grain of the Zauralochka variety;
E. food ingredient obtained by milling the milled red germinated and fermented grain soft spring wheat of the Erythrosperium variety according to the above-mentioned technology;
F. food ingredient obtained by milling germinated and fermented according to the above-described technology spring barley grain variety Chelyabinets 1.

### 2.2. Obtaining Food Ingredients from Sprouted Cereals

#### 2.2.1. Ultrasonic Treatment of Crops

In order to remove contaminants and foreign matter, the crops were prewashed in running water at 20 ± 2 °C in five repetitions.

Prior to the germination process, the crops were treated with ultrasound in a hydro-module (a system consisting of an ultrasonic generator and a 1 litre water tank with soaked stainless-steel grains). The treated grains were constantly stirred. For ultrasonic treatment, the method described by [19] was used.

Ultrasonic treatment (22 ± 1.25 kHz) was performed by means of changing power and length of time (245 W/L, intensity for 5 min). Volumetric energy density was calculated considering the thermal capacity and volume of the medium being treated (1 L). High

intensity ultrasound (20 W/cm$^2$) was applied [35–37]. An ultrasonic technological device (Volna-M UZTA-0.63/22-OM, Biysk, Russia) was used for treatment [38,39].

### 2.2.2. Crop Germination

After ultrasonic treatment, the cereals (500 g) were soaked in water at 22 ± 2 °C for 8 h (wheat grain) and 12 h (barley grain). The samples were then placed in germination trays that were kept in a temperature and humidity-controlled chamber (Seed germination cabinet SHPZ, Saratov, Russia). Germination was carried out at 22 ± 2 °C and 95 ± 3% relative humidity with humidity. Germinated wheat and barley grains were removed from the chamber when a sprout size of 1.5–2 mm was reached in more than 90% of the grains. The germination time was 16 to 24 h.

### 2.2.3. Cereal Fermentation

A commercial starter "Vivo Probio" (Kiev, Ukraine) was used as a starter microflora for the fermentation process (Kiev, Ukraine), which includes Streptococcus thermophilus, Lactobacillus delbrueckii ssp. Bulgaricus, Lactobacillus acidophilus (2 strains), Bifidobacterium lactis (2 strains), Lactobacillus casei, Lactobacillus rhamnosus, Lactobacillus paracasei, and Bifidobacterium infantis. This sourdough starter was purchased at the Chelyabinsk market. The fermentation process was carried out as recommended by the manufacturer. It has a fairly wide range of lactic acid microorganisms which, according to [40], allow a rapid increase in GABA content during fermentation of plant raw materials. For the fermentation process of germinated samples of *Triticum aestivum* L. and *Hordeum vulgare* L., the dry starter "Vivo Probio" was diluted with sterile distilled water to a concentration of $1 \times 10^7$ CFU/mL, respectively, and activated at 36 °C for 1 h. Then, 500 g of the germinated cereals were accurately weighed and mixed with 10 mL of the activated starter solution. Fermentation was then carried out at 36 °C for 18 h (according to the manufacturer's recommendations).

### 2.2.4. Drying and Grinding Sprouted Cereals

The germinated and fermented crops were dried (drying oven M 720, Binder, Baddeckenstedt, Germany) at 37 ± 2 °C for 10 h until a final moisture content of 8–10% was reached.

Food ingredients were obtained by grinding germinated and fermented cereals using a Perten 3100 laboratory mill at a fixed speed of 20,000 rpm (Perten Instruments, Stockholm, Sweden), equipped with a 0.8 mm metal mesh, followed by a 0.6 mm sieving process [41]. The grinding time was 180 s until the raw material with stable particle sizes was obtained.

### 2.2.5. Particle Size Analysis

Particle size analysis of food ingredients was carried out by laser dynamic light scattering on a Microtrac S3500, Haan, Germany (AACC 55–40.01, 2010) [42]. Samples (3 g as is/l) were dispersed in iso-propanol (100%). The 1000 mm optics was used, with a size range of 2–800 μm. The results were expressed as d(0.1), d(0.5), and d(0.9), corresponding to the maximum diameters of 10%, 50%, and 90% of the particles, respectively (in % of total volume) [43].

### 2.2.6. Physical and Chemical Analysis of Food Ingredients

In grain crops and the raw ingredients obtained, moisture content was determined using a laboratory oven set to 105 °C, relative humidity 1.1% (PE-4650, Novosibirsk, Russia) according to the AACC 14-15 method. Ash content was evaluated using a laboratory furnace model 6.7/1300 s (SNOL, Novosibirsk, Russia) according to the AACC 08-01 method. Total protein content was evaluated using a Kjeldahl model 8400 analyzer (Kjeltec, Genève, Switzerland) in accordance with AACC No. 12-46. Total fat content was assessed using a Soxchelt model PBI extractor (Buchi, Heerbrugg, Italy) in accordance with AACC No. 30-25 [44,45]. Total starch was determined by polarimetric method (GOST 10845-98) by

dissolution of starch in hot diluted solution of hydrochloric acid followed by determination of optical angle of rotation of starch solution using polarimeter POL-DISC4 (Bioevopeak, Jinan, Shandong, China). The mass fraction of mono- and disaccharides (sugars) was determined photometrically according to GOST 26176-2019 using photocalorimeter KFK-3KM (Unico-Sis, Novosibirsk, Russia). Phytic acid was determined by the optical density of the solution using a Jenway spectrophotometer (6405 UV/Vis, Dunmow, UK) at 415 nm according to GOST R ISO 30024-2012.

### 2.2.7. Antioxidant Activity (DPPH)

For the radical scavenging assay (DPPH), the radical scavenging activity of the samples was measured using 2,2-diphenyl-1-picrylhydrazyl (DPPH, (manufacturer Alfa Aesar, Karlsruhe, Germany)) [32]. The extracts (0.5 mL) were mixed with 3.6 mL of radical solution (0.025 g DPPH in 100 mL ethanol). The absorbance of the sample extract was determined using a Jenway spectrophotometer (6405 UV/Vis, Dunmow, UK) at 515 nm. Trolox (6-hydroxy-2,5,7,8-tetramethylchroman-2-carboxylic acid) (10–100 mg/L; $R^2 = 0.988$) was used as a standard, and results were expressed in $mg^{-1}$ Trolox equivalent.

### 2.2.8. Total Flavonoid Content

Determination of total flavonoid content was carried out using the procedure described by Shafii [46]. A total of 0.5 mL of sample extract was mixed with 0.1 mL of 10% (*w/v*) Ethanol aluminium chloride solution, 0.1 mL of 1 M sodium acetate, and 4.3 mL of distilled water. After 30 min in the dark, the absorbance at 415 nm was measured using a Jenway spectrophotometer (6405 UV/Vis, Dunmow, UK). Quercetin (0.01–0.5 mg $L^{-1}$; $R^2 = 0.997$) was used as a standard and the results were expressed in μg-1-equivalents of quercetin.

### 2.2.9. Determination of GABA

Germinated wheat grain (1.00 g) was ground with 6 mL of 4% acetic acid. The homogenate was left on a shaker for 1 h, in order to allow sufficient GABA release and then centrifuged at $6000 \times g$. for 15 min. The supernatant was collected and treated with 4 mL ethanol in order to remove macromolecular polymers and then centrifuged at $16,770 \times g$ for 20 min. The purified supernatant was evaporated (0.1 MPa, 45 °C) to volatilise acetic acid and ethanol. The residue was dissolved in 0.5 mL distilled water and centrifuged at $2683 \times g$ for 10 min.

The centrifuged suspension was filtered through a 0.45 μm membrane filter. A total of 100 μL of the filtered supernatant was analyzed by HPLC (Shimadzu Prominence LC-20 automated liquid chromatography system, Kyoto, Japan) with a Prodigy C 18 reverse phase column (5 μm) with an inner diameter of 4.6 × 250 mm, as described by Rossetti and Lombard (1996) [47,48]. The GABA standard solution and sample were determined by pre-column derivatisation of phenylthiocarbamyl-GABA (PTC-GABA) from PITC. The mobile phase A consisted of 70 mM sodium acetate buffer solution (pH 5.8) treated with 0.5 mL triethylamine per litre of buffer; and mobile phase B was acetonitrile. The elution system employed 92% mobile phase A and 8% mobile phase B at a flow rate of 0.5 mL/min throughout the cycle. Twenty microlitres of each sample was injected and detected at 254 nm, at a column temperature of 27 °C.

### 2.2.10. Assimilability Criterion and Growth of Infusoria in Nutrient Media

The digestibility criterion was determined by means of the biotesting technique [49] using protozoan cultures of *Tetrahymena pyriformis* (Europolitest LLC, Moscow, Russia). A pure culture of protozoa was pre-grown in nutrient medium for 48 to 96 h at 25 ± 2 °C.

Simultaneously, a suspension of crushed grain was prepared in distilled water, hydrochloric acid was added to a pH value of 3, pepsin was injected, and the suspension was extracted for 2 h; caustic soda was then injected to a pH value of 8, and pancreatin and extraction was carried out for 2 h. Hydrochloric acid was added to pH 6, live biomaterial was fed by diluting the working culture of infusoria in the extract, samples were placed in a

multiwell plate of BioLaT-3.2, and the number of infusoria in the wells was counted before the experiment and after 24 h; through their ratio, the digestibility criterion was calculated.

Assimilability criterion (*Ac*) was calculated according to the formula:

$$A_c = \frac{K_{gs}}{Kgw} \times 100 \tag{1}$$

where *Kgs* is the coefficient of culture growth in the sample; *Kgw* culture growth factor in distilled water (control).

### 2.2.11. Statistical Analyses

The research was carried out in triplicate. Cereal germination and food ingredient production were carried out under the same conditions to ensure the accuracy of the results. Results were expressed as mean values of five replications ± standard deviation. Probability values of $p \leq 0.05$ were taken to indicate statistical significance. Data were analyzed by one-way ANOVA followed by Duncan's multiple range tests using IBM SPSS Statistics version 28.0 (IBM, Armonk, NY, USA) proposed by Assaad et al. [50].

## 3. Results and Discussion

### 3.1. Average Particle Size Distribution of Food Ingredient Samples

Milling is one of the most important processes for the processing of germinated crops. It is the accuracy of this process that determines how widely the raw ingredients can be used in food production.

The main grinding criterion used in the processing of grain crops is to obtain flour particles of a fixed size [51]. The processes that occur during germination and fermentation affect biochemical changes as well as structural changes in the grain. As a result, the strength properties of the grain change, especially the hardness, on which the energy required for milling depends, decreases. Germinated and fermented cereals should not be milled by classical milling because the high plasticity of the grain reduces the yield of the food ingredient [52].

The process of preparing germinated and fermented cereals for milling is simple because the grain mass is cleaned prior to these operations. For dry milling, the moisture content of the germinated grain must be reduced to an optimum level (no higher than 14%) [53]. The results of the milling process are also highly dependent on the type of mill used. In our work, a Perten 3100 (Perten Instruments, Stockholm, Sweden) laboratory mill was used, with a fixed speed of 20,000 revolutions per minute. The resulting food ingredients were sieved through a 0.6 mm sieve [41]. The grinding time was 180 s until a flour with stable particle size was obtained. Characteristics such as particle size distribution were used to characterize the milling process (Figure 1).

Particle size distribution is a very important factor from a technological point of view, since it influences the properties of the food ingredients and their interaction in the food matrix, affecting their quality [54]. The results showed that the milled material obtained from germinated and fermented wheat grain (D and E) was characterized mainly by more uniform distribution, low proportion of fine particles (4.62–104.60 µm ($p \leq 0.05$)), and large particles (418.60–592.00 µm ($p \leq 0.05$)). A sample of germinated and fermented barley grain (F) had the most uniform particle size distribution (31.11–248.90 µm ($p \leq 0.05$)). The granulometric composition of the obtained food ingredients is shown in Table 1.

The particle size d(0.5) for the germinated and fermented samples decreased from 296.00 to 209.30 µm ($p \leq 0.05$) for A and D; from 248.91 to 148.00 µm ($p \leq 0.05$) for B and E; and from 296.00 to 124.50 µm ($p \leq 0.05$) for C and F. The reduction in grain hardness resulting from germination and fermentation resulted in a more uniform particle size distribution which will allow the resulting raw ingredients to be more uniformly distributed in the food matrix.

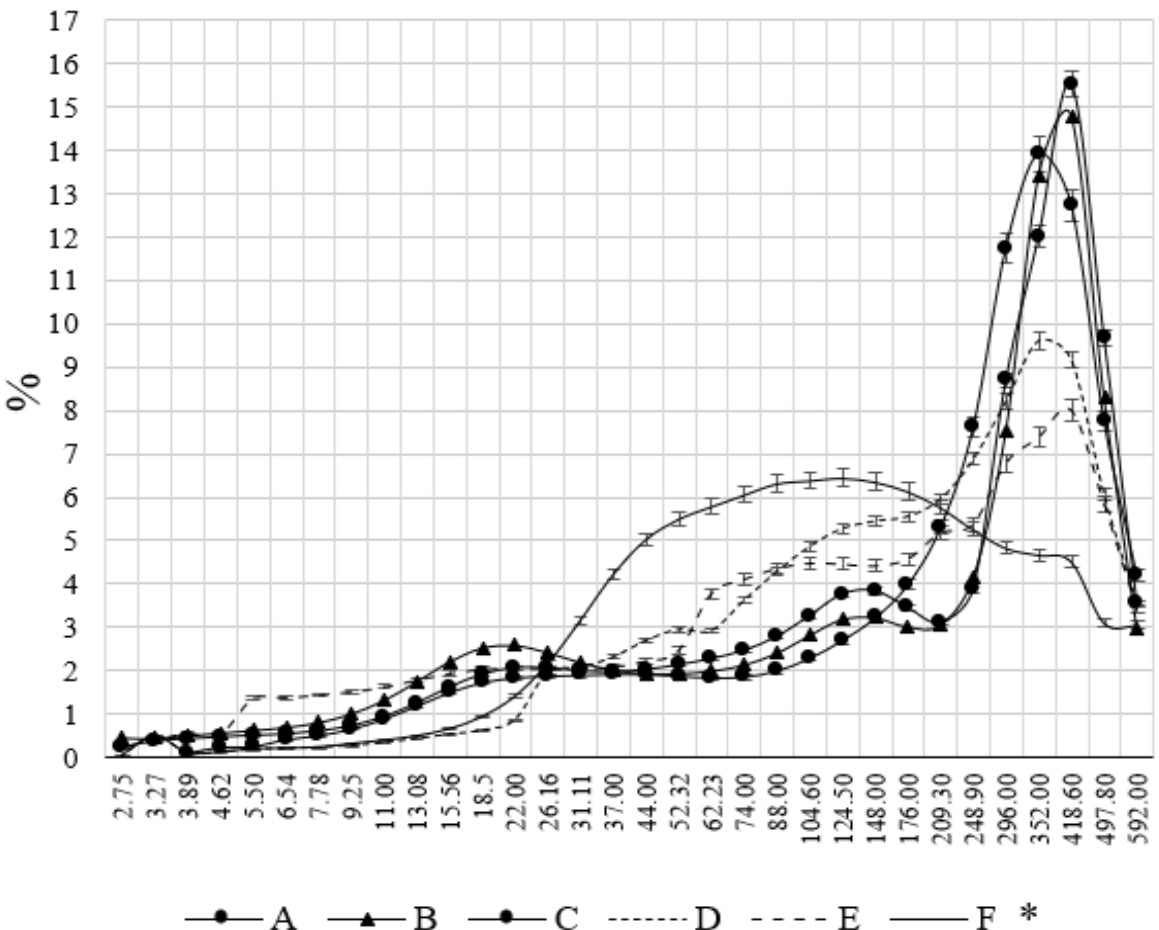

**Figure 1.** Particle size distribution profile of food ingredients. * Abbreviations: A—food ingredient obtained by milling whole grain wheat of Zauralochka; B—food ingredient obtained by milling whole grain wheat of Erythrosperium; C—food ingredient obtained by milling whole spring barley Chelyabinets 1; D—food ingredient obtained by milling germinated and fermented according grain variety Zauralochka; E—food ingredient obtained by milling germinated and fermented according grain variety Erythrosperium; F—food ingredient obtained by milling germinated and fermented according grain variety Chelyabinets 1. (Significant differences ($p < 0.05$) in comparison to reference).

**Table 1.** Particle Size Distribution of Food Ingredients *.

| Indicator Name | A * | B * | C * | D * | E * | F * |
|---|---|---|---|---|---|---|
| d(0.1) (μm) | 22.00 [a] | 15.56 [b] | 18.50 [b] | 37.00 [a] | 13.08 [a] | 13.08 [c] |
| d(0.5) (μm) | 296.00 [a] | 248.91 [c] | 296.00 [c] | 209.30 [a] | 148.00 [a] | 124.50 [b] |
| d(0.9) (μm) | 497.80 [a] | 497.80 [b] | 497.81 [c] | 497.82 [a] | 497.83 [a] | 418.60 [b] |

* Abbreviations: A—food ingredient obtained by milling whole grain wheat of Zauralochka; B—food ingredient obtained by milling whole grain wheat of Erythrosperium; C—food ingredient obtained by milling whole spring barley Chelyabinets 1; D—food ingredient obtained by milling germinated and fermented according grain variety Zauralochka; E—food ingredient obtained by milling germinated and fermented according grain variety Erythrosperium; F—food ingredient obtained by milling germinated and fermented according grain variety Chelyabinets 1. The values with different letters (a–c) indicate significant differences at $p < 0.05$ using Duncan's multiple range test.

### 3.2. Physico-Chemical Analysis of Food Ingredients

The results of the chemical composition are shown in Table 2.

**Table 2.** Chemical composition of food ingredients *.

| Name of Indicator | A * | B * | C * | D * | E * | F * |
|---|---|---|---|---|---|---|
| Crude protein (% d.b.) | 11.6 ± 0.3 [a] | 14.8 ± 0.3 [b] | 13.2 ± 0.3 [a] | 13.9 ± 0.3 [b] | 16.3 ± 0.3 [b] | 13.9 ± 0.3 [b] |
| Lipids (% d.b.) | 1.4 ± 0.2 [bc] | 1.8 ± 0.2 [b] | 1.9 ± 0.2 [c] | 1.9 ± 0.2 [a] | 2.1 ± 0.2 [b] | 2.3 ± 0.2 [b] |
| Mass fraction of starch, % | 60.3 ± 0.6 [a] | 58.9 ± 0.6 [a] | 61.6 ± 0.5 [a] | 41.6 ± 0.5 [a] | 42.4 ± 0.5 [c] | 41.8 ± 0.5 [c] |
| Mass fraction of mono- and disaccharides, % | 1.2 ± 0.4 [c] | 1.9 ± 0.4 [a] | 1.6 ± 0.4 [c] | 3.6 ± 0.4 [bc] | 4.2 ± 0.4 [c] | 4.6 ± 0.4 [bc] |
| Phytic acid content, g/100 g dry matter | 2.3 ± 0.2 [a] | 2.5 ± 0.2 [a] | 2.8 ± 0.2 [a] | 1.7 ± 0.2 [a] | 1.4 ± 0.2 [a] | 1.2 ± 0.2 [a] |

The values with different letters (a–c) indicate significant differences at $p < 0.05$ using Duncan's multiple range test. * Abbreviations: A—food ingredient obtained by milling whole grain wheat of Zauralochka; B—food ingredient obtained by milling whole grain wheat of Erythrosperium; C—food ingredient obtained by milling whole spring barley Chelyabinets 1; D—food ingredient obtained by milling germinated and fermented according grain variety Zauralochka; E—food ingredient obtained by milling germinated and fermented according grain variety Erythrosperium; F—food ingredient obtained by milling germinated and fermented according grain variety Chelyabinets 1.

The increase in the Lipids can be attributed to the increase in free lipids provided by the germination and fermentation process. The mass fraction of starch decreases significantly during the germination and fermentation process of crops. Regarding protein content relative to control samples, samples D, E, and F after the germination and fermentation processes showed an increase of 13.3%; 10.1%, and 5.3% ($p \leq 0.05$), respectively. These data agree with Hadnadev [55] who reported similar results where the wheat most exposed to high temperatures and precipitation prior to harvest presented the highest values of wet gluten content. The changes observed in gluten formation after germination and fermentation may be due to the formation of phenolic compounds capable of binding sulfides available to form disulfide bridges [56,57].

Cereal germination and fermentation occur under the influence of moisture and the microorganisms used, initiating numerous physiological and biochemical processes to support these processes. Many enzymes are activated, and some nutrients and phytochemicals are released from the germ, endosperm, and shell parts. A group of substances with a small molecular weight account for the antioxidant properties of germinated and fermented food ingredients, among which can be identified, carotenoids, tocopherols, flavonoids, phenolic acids, and other substances [58].

Phytic acid is a serious anti-nutrient for the human body. Even small amounts almost completely interfere with the absorption of essential nutrients such as calcium, iron, zinc, magnesium, manganese, phosphorus, as well as cyanocobalamin (vitamin B12). Because of its molecular structure, which is a complex ester of the cyclic six-atom polyalcohol myo-inositol and six orthophosphoric acid residues, phytic acid binds most substances with a positive oxidation degree, of which metals form the basis.

Two processes can be used to reduce phytic acid content in cereals: germination and fermentation. Germination reduces the phytic acid content, but fermentation does this more effectively. A combination of germination and fermentation is the most effective way of reducing the phytic acid content, relative to the final content of useful nutrients in the finished product.

Experimental data indicates a pronounced effect of the germination and fermentation process on the phytic acid content. In samples D, E, and F, the phytic acid content was reduced by 26%, 32%, and 57% ($p \leq 0.05$), respectively, significantly increasing the iron bioavailability of the developed food ingredients.

Changes in total antioxidant activity and flavonoid content during germination and fermentation of cereals are shown in Figure 2.

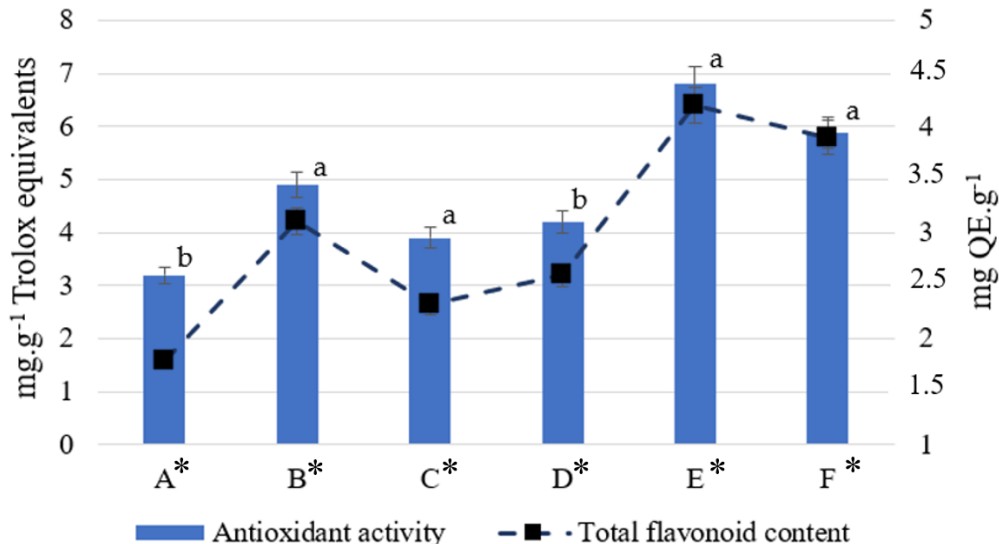

**Figure 2.** Changes in total antioxidant activity and flavonoid content during germination and fermentation of cereals. The bars with different letters (a,b) indicate significant differences at $p < 0.05$ using Duncan's multiple range test. * Abbreviations: A—food ingredient obtained by milling whole grain wheat of Zauralochka; B—food ingredient obtained by milling whole grain wheat of Erythrosperium; C—food ingredient obtained by milling whole spring barley Chelyabinets 1; D—food ingredient obtained by milling germinated and fermented according grain variety Zauralochka; E—food ingredient obtained by milling germinated and fermented according grain variety Erythrosperium; F—food ingredient obtained by milling germinated and fermented according grain variety Chelyabinets 1.

The data presented in Figure 2 indicate a significant increase in flavonoids and an increase in total antioxidant activity during germination and fermentation of crops. It should be noted that ultrasound intensification of the germination process caused significant changes in these indicators, which was proved earlier and presented in [36]. Thus, the value of total antioxidant activity increased in samples D, E, and F by 31, 38, and 51% ($p \leq 0.05$) on average, while the amount of flavonoids increased by 35, 45, and 68% ($p \leq 0.05$) relative to samples A, B, and C.

Wheat and barley are rich in biologically active compounds such as flavonoids, phenolic acids, lignans, stilbenes, and diterpenes. These compounds have different molecular weights and chemical structures and are widely distributed in plants in free and bound forms [59]. Phenolic compounds are known for their antioxidant activity; they play a key role in the treatment and prevention of several diseases (such as cardiovascular and neurodegenerative diseases) and cancer [60]. In our experiment, the highest concentration of polyphenolic compounds and flavonoids was found in food ingredients obtained by milling germinated and fermented according to the above-described technology soft spring red-grain wheat grain of the variety Erythrosperium (4.2 mg QE·g$^{-1}$).

The effect of microbial fermentation on flavonoid content in fermented grain-based products has been reported in numerous studies [61]. Conducting a fermentation process increases the solubility and extractability of flavonoids, making them more bioavailable. This process depends on the strain and specificity of the enzymatic activity of bacteria. Therefore, a complex inoculum was used to obtain the greatest effect.

Another important substance accumulated during germination and fermentation of crops is γ-aminobutyric acid. This effect is particularly pronounced when the process is activated using ultrasound exposure [62,63]. γ-aminobutyric acid is a non-protein amino acid mainly produced by the γ-decarboxylation of 1-glutamic acid, which is catalysed by the enzyme glutamate decarboxylase [64]. γ-aminobutyric acid functions as a major

inhibitory neurotransmitter in humans [65]. To date, there has been increased interest in the use of γ-aminobutyric acid as a biologically active component of plant foods.

HPLC analysis showed that every sample of both control and germinated and fermented cereal samples contained γ-aminobutyric acid (Figure 3). The γ-aminobutyric acid content of samples D, E, and F was 4.7, 3.1, and 3.2 times ($p \leq 0.05$) (compared to the control sample), which would allow the production of food products with a pronounced preventive effect.

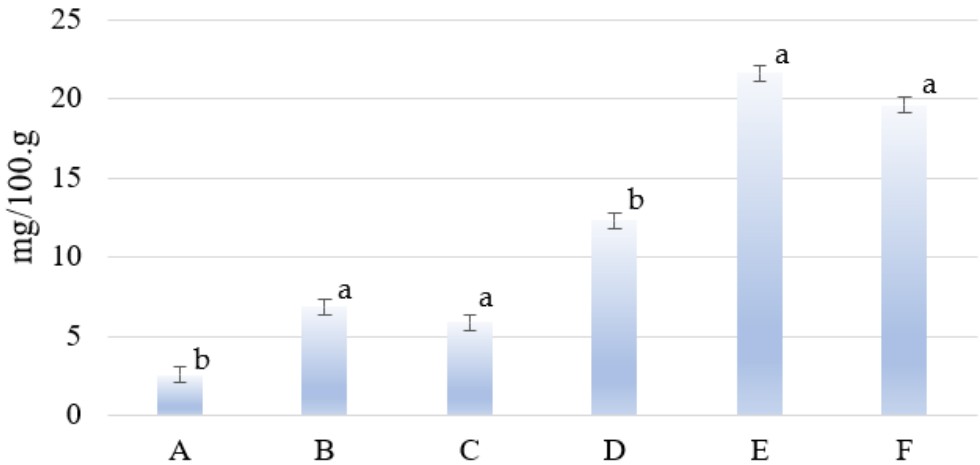

**Figure 3.** Changes in GABA during germination and fermentation of cereals. bbreviations: A—food ingredient obtained by milling whole grain wheat of Zauralochka; B—food ingredient obtained by milling whole grain wheat of Erythrosperium; C—food ingredient obtained by milling whole spring barley Chelyabinets 1; D—food ingredient obtained by milling germinated and fermented according grain variety Zauralochka; E—food ingredient obtained by milling germinated and fermented according grain variety Erythrosperium; F—food ingredient obtained by milling germinated and fermented according grain variety Chelyabinets 1. The bars with different letters (a,b) indicate significant differences at $p < 0.05$ using Duncan's multiple range test.

Using a combination of ultrasound exposure before the germination process and subsequent fermentation of cereal crops in the technology of food ingredients allows increasing the GABA content by 8–20% ($p \leq 0.05$) compared to the previously obtained results [19].

Previously, researchers reported that lactic acid bacteria have an excellent ability to synthesize GABA. Their ability varies greatly between species and strains [66]. In this study, a complex inoculum consisting of Lactobacillus casei and Lactobacillus rhamnosus was used. As reported by the authors [13,60], these microorganisms have a maximum ability to bioconvert GABA, which accounts for the above result.

The germination process is known to activate the action of enzymes, thus helping to increase the digestibility of the grain.

The use of Tetrahymena pyriformis test organisms is a promising research method that is based on recording the growth of the culture for a given time, which depends directly on the bioavailability of nutrients and their ratio in the extracts of the products analysed. The results of the digestibility criterion are shown in Figure 4.

Cereal germination and fermentation processes increase the digestibility of food ingredients. Using nutrient media of D, E, and F samples increases growth of Tetrahymena pyriformis infusoria by 1.9, 1.7, and 1.4 ($p \leq 0.05$) times within 24 h, which, in turn, increases the digestibility criteria by an average of 93.9, 71.5, and 42.4% ($p \leq 0.05$), respectively. This is probably because the germination and fermentation regimes which we used stimulate the breakdown of protein and starch by proteolytic and amylolytic enzymes.

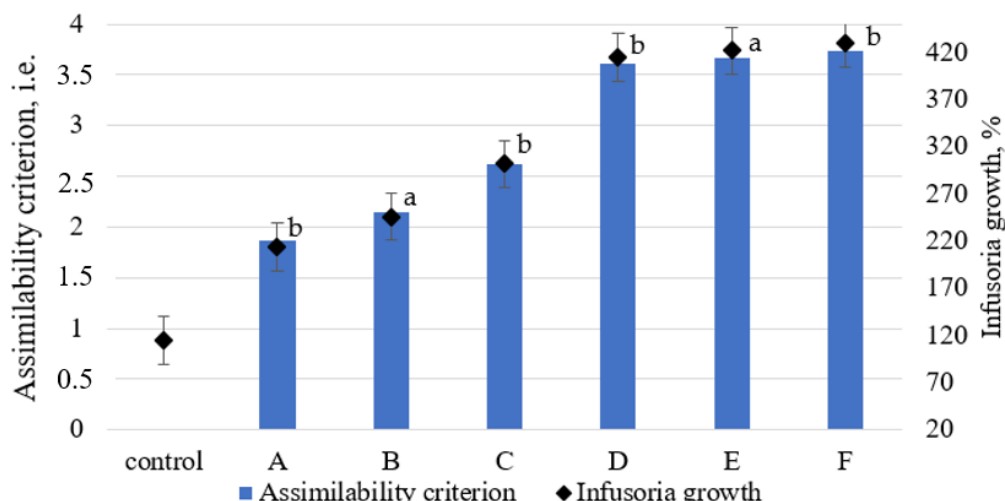

**Figure 4.** Results of digestibility criterion and growth of infusoria in nutrient media based on control and fermented cereals: A—food ingredient obtained by milling whole grain wheat of Zauralochka; B—food ingredient obtained by milling whole grain wheat of Erythrosperium; C—food ingredient obtained by milling whole spring barley Chelyabinets 1; D—food ingredient obtained by milling germinated and fermented according grain variety Zauralochka; E—food ingredient obtained by milling germinated and fermented according grain variety Erythrosperium; F—food ingredient obtained by milling germinated and fermented according grain variety Chelyabinets 1. The bars with different letters (a,b) indicate significant differences at $p < 0.05$ using Duncan's multiple range test.

## 4. Conclusions

The technology of germination of cereal crops followed by fermentation used in this study could be one of the effective technologies producing new generation food ingredients. Enzymatic processes play a key role in the production of whole-grain food ingredients for food due to their improved functional and antioxidant properties. We found that the food ingredients obtained from sprouted and fermented raw materials had a more aligned particle size distribution, with particle sizes d(0.5) of 209.30, 148.00, and 124.50 µm for D, E, and F, respectively. Carrying out the fermentation process further increased both the flavonoid content and the antioxidant activity. Thus, the value of total antioxidant activity increased in D, E, and F samples by 31, 38, and 51% on average, and the amount of flavonoids by 35, 45, and 68% ($p \leq 0.05$) (compared to the control sample).

As a neurotransmitter in the body, GABA is worthy of attention as a functional food material because it can normalize blood pressure and improve liver function and alcohol metabolism when consumed regularly. It is difficult to express the functionality of GABA present in plant foods by natural ingestion alone, as it is found there in trace amounts.

In this study, we proposed a method to increase the content of GABA by combining the processes of germination with ultrasound treatment and fermentation with a complex leavening of cereal crops. The use of this technique increased the GABA content of D, E, and F samples by 4.7, 3.1, and 3.2 times ($p \leq 0.05$) (compared to the control sample). Also, this technology enhances the digestibility of the resulting food ingredients.

Future studies are planned which will expand to more varieties of cereal crops and the possibility of placing them in the matrix of food products such as bread, vegetable, milk, and snacks. In order to fully utilize the benefits of fermentation in the production of food ingredients, including those with antioxidant properties, commercial microbial strains must be carefully selected with metabolomics principles in mind. In addition, the optimization of the fermentation process parameters based on mathematical modeling methods is necessary in order to comprehensively improve the quality and nutritional value of food products.

**Author Contributions:** Conceptualization, N.N.; Methodology, N.N., R.F. and I.K.; Investigation, N.N., R.F. and N.P.; Data curation and analysis, E.V., R.F. and A.R.; Supervision, N.N.; Writing— original draft R.F., A.A., R.M., V.V.A., E.V. and I.K.; Writing—article and editing R.F., I.K., N.N. and N.P. All authors have read and agreed to the published version of the manuscript.

**Funding:** This research was partially supported by a grant from the Russian Science Foundation 23-26-00290.

**Institutional Review Board Statement:** Not applicable.

**Informed Consent Statement:** Not applicable.

**Data Availability Statement:** Data will be made available on request to the corresponding author.

**Acknowledgments:** We would like to thank the managers of the Nanotechnology Research and Education Center of South Ural State University (Chelyabinsk, Russia) for their technical support during this work.

**Conflicts of Interest:** The authors declare no conflict of interest. The funders had no role in the design of the study; in the collection, analyses, or interpretation of data; in the writing of the manuscript, or in the decision to publish the results.

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
