# Peer review of "Effect of a Combination of Ultrasonic Germination and Fermentation Processes on the Antioxidant Activity and γ-Aminobutyric Acid Content of Food Ingredients"

_fermentation, doi:10.3390/fermentation9030246_

Round 1

Reviewer 1 Report

General comments:

ü  The English language must revise

ü  The editorial issues are persistent

ü  Typographical issues must consider

ü  Once check for the vocabulary and spellings in the manuscript.

ü  The title of the manuscript should be revised as it is showing as the statement and there is not mentioned of the ultra-sonication

Abstract:

ü  Must revise

ü  The flow is very poor

ü  Need to represent the study Methods and the results and conclusions

ü  Why select wheat? Why not another crop? should mention in the abstract

Introduction:

ü  Very poor,

ü  Limited information given

ü  Must give the Gap, Previous studies reported

ü  Must talk about the germination of wheat

ü  The germination of the wheat should be mentioned in the introduction

ü  Must mention the fermentation information

ü  Must give the issues on what the application of your research or the wheat powder produced?

Materials and Methods:

ü  Section 2.1 What is the logic behind the selection of these samples?

ü  Section 2.2.3 Must give why selected specific microbes only. How many days fermented?

ü  Section 2.2.4 Must give the exact temperature

ü  Section 2.2.6. looks like the proximate composite

ü  Section 2.2.11. Data analysis? Need to change the title

Results and discussion:

ü  Line 193 to 206 looks like an introduction, must revise this

ü  Replace with “.” In place of “,” in table 1

ü  The discussion on the antioxidant properties must discuss very deeply.

ü  Must discuss the GABA also in divergent

ü  I hope some analyzed properties are not discussed, check once

ü  Better give intended applications of the products

Conclusions:

ü  Must revise the properties

ü  Not analyzed the mineral so, should give conclusions what analyzed

Author Response

Dear Reviewer,

We thank you for your careful consideration of our manuscript and your constructive comments and suggestions for improving the material presented.

Also in accordance with your comments, we have expanded the Abstract, Keywords, Introduction, Materials, Methods, Results, Discussion and Conclusions part и in our manuscript.

Absolutely all individual comments have been taken into account and corrected in our new version of the manuscript, in particular

General comments:

ü  The English language must revise

ü  The editorial issues are persistent

ü  Typographical issues must consider

ü  Once check for the vocabulary and spellings in the manuscript.

The manuscript material has been thoroughly revised, improved, and further proofread (we attach the appropriate report)

ü  The title of the manuscript should be revised as it is showing as the statement and there is not mentioned of the ultra-sonication

The title of the manuscript has been changed to be more specific about the content of the article: Effect of a combination of ultrasonic germination and fermen-tation processes on the antioxidant activity and γ-aminobutyric acid content of food ingredients

Abstract:

ü  Must revise

ü  The flow is very poor

ü  Need to represent the study Methods and the results and conclusions

ü  Why select wheat? Why not another crop? should mention in the abstract

The abstract materials are revised as much as possible, the research methods, results and conclusions are presented. The choice of research objects is justified. The material is presented on lines: 15-40.

Introduction:

ü  Very poor,

ü  Limited information given

The abstract materials are maximally revised and greatly improved, lines 48-102

ü  Must give the Gap, Previous studies reported

Results from previous studies are indicated, lines 69-72, as well as the relevance of this study, lines 96-99

ü  Must talk about the germination of wheat

Describes the process of germination of crops, lines 48-52

ü  The germination of the wheat should be mentioned in the introduction

Describes changes in the chemical composition of crops during germination, lines 63-87

ü  Must mention the fermentation information

Information on the fermentation of crops, lines 88-95

ü  Must give the issues on what the application of your research or the wheat powder produced?

Revised and clarified the purpose of this research, lines 100-103

Materials and Methods:

ü  Section 2.1 What is the logic behind the selection of these samples?

A rationale for the choice of samples is given, lines 105-107

ü  Section 2.2.3 Must give why selected specific microbes only. How many days fermented?

Fermentation conditions are clarified, the choice of complex starter is justified, lines 161-174

ü  Section 2.2.4 Must give the exact temperature

Specifies the exact temperature, line 177

ü  Section 2.2.6. looks like the proximate composite

Methods clarified and detailed, lines 189-203

ü  Section 2.2.11. Data analysis? Need to change the title

Title changed to "Statistical Analyses"

Results and discussion:

ü  Line 193 to 206 looks like an introduction, must revise this

Sample designation moved to section 2.1, lines 125-138

ü  Replace with “.” In place of “,” in table 1

In Table 1, "," is replaced by "."

ü  The discussion on the antioxidant properties must discuss very deeply.

A discussion of antioxidant properties is presented,lines 361-376

ü  Must discuss the GABA also in divergent

Presented discussion of GABA, lines 402-410

ü  I hope some analyzed properties are not discussed, check once

A discussion of all the experimental data is given

ü  Better give intended applications of the products

Specifies the possibility of using food ingredients in products, lines 451-458

Conclusions:

ü  Must revise the properties

ü  Not analyzed the mineral so, should give conclusions what analyzed

The conclusion is detailed, an analysis of the data obtained is presented, conclusions are drawn, lines 432-458

Reviewer 2 Report

The previous work by the authors (ref. 13) is on a similar topic. One difference is the fermentation step used in the submitted work (in addition to ultrasound and germination used previously). However, particularly this step is not well described. It was based on using a poorly defined commercial preparation. A clear specification of the crops varieties is also missing (no references).

The advances against the previous work are not clearly discussed. In addition, a number of word-to-word citations form this work have been found.

Author Response

Dear Reviewer,

On behalf of my co-authors, we thank you very much for allowing us to revise our manuscript. We deeply appreciate the reviewer for their positive and constructive comments and suggestions on our manuscript entitled “Effect of a combination of ultrasonic germination and fermen-tation processes on the antioxidant activity and γ-aminobutyric acid content of food ingredients”.

Those comments are all valuable and very helpful for revising and improving our paper, as well as the essential guiding significance to our research. We carefully revised this manuscript according to the comments of editors and reviewers.

We appreciate for Reviewer warm work earnestly. Also in accordance with your comments, we have expanded the Abstract, Keywords, Introduction, Materials, Methods, Results, Discussion and Conclusions part и in our manuscript.

Please find the revised version attached, which we would like to submit for your kind consideration. Here the response to reviewers and changes in details are all listed in this letter.

The previous work by the authors (ref. 13) is on a similar topic. One difference is the fermentation step used in the submitted work (in addition to ultrasound and germination used previously). However, particularly this step is not well described. It was based on using a poorly defined commercial preparation.

Perhaps you meant the article (ref. 25) (Naumenko, N.; Potoroko, I.; Kalinina, I. Stimulation of antioxidant activity and γ-aminobutyric acid synthesis in germinated wheat grain Triticum aestivum L. by ultrasound: Increasing the nutritional value of the product. Ultrasonics Sonochemistry 2022, 86, 106000. 10.1016/j.ultsonch.2022.106000), as with the above article (ref. 13) (Kalinina, I.; Fatkullin, R.; Naumenko, N.; Ruskina, A.; Popova, N.; Naumenko, E. Increasing the Efficiency of Taxifolin Encapsulation in Saccharomyces cerevisiae Yeast Cells Based on Ultrasonic Microstructuring. Fermentation 2022, 8, 378. https://doi.org/10.3390/fermentation8080378) elements of similarity are completely absent.

Two other varieties of wheat (grain of soft spring white wheat, variety Zauralochka and grain of soft red spring wheat, variety Erythrosperium, 2022) and a sample of barley (spring barley grain, variety Chelyabinets 1) are used in the article.

The conduct of the fermentation process of the objects is considerably specified, the choice of a complex starter, its composition, lines 161-174 is justified.

In the Results and Discussion, emphasis is placed on the increase in antioxidant activity, flavonoid and γ-aminobutyric acid content during fermentation, as well as digestibility criterion and growth of infusoria in nutrient media of the obtained food ingredients.

Detailed data on the use of a complex fermentative starter are given. Distinctive features proving scientific novelty are presented in the table. (Attached file)

Thus, the team of authors believes that the research presented have scientific novelty, and the developed technology of high efficiency and practical relevance.

A clear specification of the crops varieties is also missing (no references).

References made to the specification of crops, lines 108-116

The advances against the previous work are not clearly discussed. In addition, a number of word-to-word citations form this work have been found.

The material of the article is revised in detail, the distinctive features of the results are presented in comparison with previous studies.

Reviewer 3 Report

Hi dear

This article "Germination and fermentation as a way to create food ingredients with increased nutritional value” was revised and has a novelty and I recommend it for publication after consideration of the following comments.

·       Ultrasound, germination and fermentation conditions applied in the research should be stated in the abstract.

·       The type of statistical design used in this research should be mentioned in the abstract.

·       You ought to change the keywords because they are repeated in the title.

·       Line 56-58: please include the treatments and response factors applied in the study.

·       Line 45-47: Does ultrasound directly lead to such events or indirectly? Please be very knowledgeable in this regard.

·       Line 48-51: please explain more about γ-aminobutyric acid background.

·                 Please write materials as Company Name (City, Country), especially for chemical analysis assessment which used in the study.

·       Line 114-115: the average particle size of food ingredients? This sentence is very vague…. Please express by detail.

·       Please express “Assimilability criterion” in materials and methods section.

·       The way of expressing the method of measuring macronutrients and other parameters has a scientific flaw. Please take help from the following article for the correct way of expressing it, so that the standard number of the working method should be clearly stated (https://doi.org/10.1590/fst.60820).

·       Line 160-166: please say how did you calculate ratio the digestibility criterion? Please include the formula.

·       Line 167- 171: please cite and use the above citation for better explain the statistical analysis.

·       Please include the milling conditions and criteria of wheat grain geminated and consequently fermentation.

·       Fig 1: would you please include the expressions include: d10, d50, d90 D3,4 and D3,2 which they are the important factors in particle size distribution.

·       All Tables and figures ought to statistical analysed: The alphabetical statistical letters for the means should all be modified such that the greatest number has the letter a and as the numbers go lower, letters b, c etc.

·       Discussion text must grammar improve and in some cases it is very weak and maybe there is no discussion at all.

·       Conclusion is very general, try to make it more scientific, comprehensive and concise in detail, especially.

References: It is OK.

The article has many flaws in express and concept of English, it is suggested to be revised in a scientific and native way.

Author Response

Dear Reviewer,

We thank you for your careful consideration of our manuscript and your constructive comments and suggestions for improving the material presented.

Also in accordance with your comments, we have expanded the Abstract, Keywords, Introduction, Materials, Methods, Results, Discussion and Conclusions part и in our manuscript.

Absolutely all individual comments have been taken into account and corrected in our new version of the manuscript, in particular

Ultrasound, germination and fermentation conditions applied in the research should be stated in the abstract.

The abstract reflects the conditions of ultrasonic exposure, germination and fermentation used in the study, lines 26-30

The type of statistical design used in this research should be mentioned in the abstract.

The abstract indicates the type of statistical plan used in this study, lines 36-37

You ought to change the keywords because they are repeated in the title.

Changed keywords: grain crops, flavonoids, complex starter, nutritional enhancement, whole grains

Line 56-58: please include the treatments and response factors applied in the study.

The goal is specified: The aim of this study was to study the combination of the processes of germination with ultrasound treatment and fermentation of complex starter cereal crops on the anti-oxidant activity and GABA content of the obtained raw ingredients with the possibility of using them in the matrix of food products.

Line 45-47: Does ultrasound directly lead to such events or indirectly? Please be very knowledgeable in this regard.

Describes the effects of ultrasonic exposure, lines 63-77

Line 48-51: please explain more about γ-aminobutyric acid background.

Entered information on the action of γ-aminobutyric acid, lines 78-94

Please write materials as Company Name (City, Country), especially for chemical analysis assessment which used in the study.

Corrected

Line 114-115: the average particle size of food ingredients? This sentence is very vague…. Please express by detail.

Corrected, lines 183-188

Please express “Assimilability criterion” in materials and methods section.

Corrected, lines 238-252

The way of expressing the method of measuring macronutrients and other parameters has a scientific flaw. Please take help from the following article for the correct way of expressing it, so that the standard number of the working method should be clearly stated (https://doi.org/10.1590/fst.60820).

Corrected, lines 190-203

Line 160-166: please say how did you calculate ratio the digestibility criterion? Please include the formula.

Corrected, lines 250-252

Line 167- 171: please cite and use the above citation for better explain the statistical analysis.

Corrected, lines 253-259

Please include the milling conditions and criteria of wheat grain geminated and consequently fermentation.

Corrected, lines 121-124, 176-181

Fig 1: would you please include the expressions include: d10, d50, d90 D3,4 and D3,2 which they are the important factors in particle size distribution.

Table 1 was added with data of d10, d50, d90 samples of received food ingredients, lines 297-309

All Tables and figures ought to statistical analysed: The alphabetical statistical letters for the means should all be modified such that the greatest number has the letter a and as the numbers go lower, letters b, c etc.

Corrected

Discussion text must grammar improve and in some cases it is very weak and maybe there is no discussion at all.

Corrected

Conclusion is very general, try to make it more scientific, comprehensive and concise in detail, especially.

The conclusion is detailed, an analysis of the data obtained is presented, conclusions are drawn, lines 432-457

References: It is OK.

The article has many flaws in express and concept of English, it is suggested to be revised in a scientific and native way.

The manuscript material has been thoroughly revised, improved, and further proofread (we attach the appropriate report)

Round 2

Reviewer 1 Report

All the comments are handled well and revised accordingly.